# Global Economic and Diet Transitions Drive Latin American and Caribbean Forest Change during the First Decade of the Century: A Multi-Scale Analysis of Socioeconomic, Demographic, and Environmental Drivers of Local Forest Cover Change

David López-Carr [1,*,†], Sadie J. Ryan [2,3,4,†] and Matthew L. Clark [5]

1 Department of Geography, University of California at Santa Barbara, Santa Barbara, CA 93106, USA
2 Department of Geography, University of Florida, Gainesville, FL 32601, USA; sjryan@ufl.edu
3 Emerging Pathogens Institute, University of Florida, Gainesville, FL 32601, USA
4 School of Life Sciences, College of Agriculture, Science, and Engineering, University of KwaZulu-Natal, Durban 4000, South Africa
5 Center for Interdisciplinary Geospatial Analysis, Department of Geography, Environment, and Planning, Sonoma State University, Rohnert Park, CA 94928, USA; matthew.clark@sonoma.edu
* Correspondence: davidlopezcarr@ucsb.edu
† These authors contributed equally to this work.

**Abstract:** Latin America and the Caribbean (LAC) contain more tropical high-biodiversity forest than the remaining areas of the planet combined, yet experienced more than a third of global deforestation during the first decade of the 21st century. While drivers of forest change occur at multiple scales, we examined forest change at the municipal and national scales integrated with global processes such as capital, commodity, and labor flows. We modeled multi-scale socioeconomic, demographic, and environmental drivers of local forest cover change. Consistent with LAC's global leadership in soy and beef exports, primarily to China, Russia, the US, and the EU, national-level beef and soy production were the primary land use drivers of decreased forest cover. National level gross domestic product (GDP), migrant worker remittances and foreign investment, along with municipal-level temperature and area, were also significantly related to reduced forest cover. This challenges forest transition frameworks, which theorize that rising GDP and intensified agricultural production should be increasingly associated with forest regrowth. Instead, LAC forest change was linked to local, national, and global demographic, dietary and economic transitions, resulting in massive net forest cover loss. This suggests an urgent need to reconcile forest conservation with mounting global demand for animal protein.

**Keywords:** economic transition; diet transition; forest change; Latin America; Caribbean; deforestation

## 1. Introduction

Dramatic yet heterogeneous shifts in forest cover occurred across Latin America and the Caribbean (LAC) during the first decade of the 21st century [1–3]. Prior to this, between 1980 and 2000, more than 55% of new agricultural land in the tropics was created from previously intact forest, and another 28% from degraded or disturbed forest [4]. LAC lost 34% (179,405km$^2$) of the 521,080 km$^2$ of global forest cover eliminated during 2001–2010 [2]. This high rate of loss in certain areas, such as the Amazon, continued in the following decade [5–7]. Despite overwhelming net loss, LAC's 2001–2010 forest change was bidirectional and heterogeneous (Figure 1), suggesting distinct regional drivers of deforestation and reforestation.

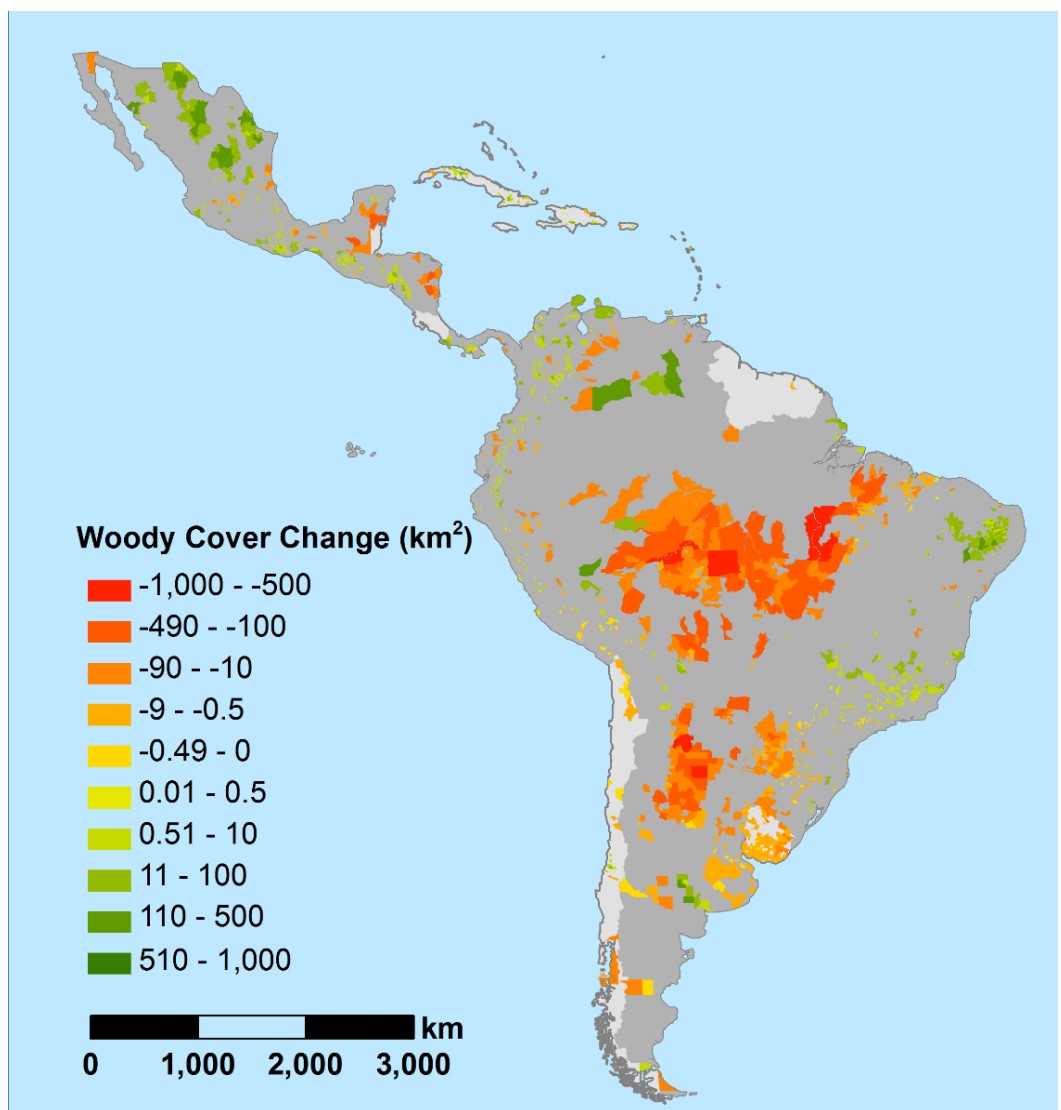

**Figure 1.** Estimated annual rate of woody cover change (in km$^2$) in LAC from years 2001 to 2010.

This is shown in the 2513 municipalities for which a significant change took place. Darker grey area denotes the 14 countries included in this analysis (92% of LAC land area).

According to the Forest Transition (FT) model, countries in early development stages undergo agricultural expansion through deforestation, followed by reforestation, eventually achieving relative forest cover stability [1,8]. Global and regional economic, demographic, and diet transitions pressure the LAC landscape through two distinct FT pathways. More developed countries (MDCs), such as Mexico and Brazil, have approached replacement level fertility, are largely urban, and have converted much of the most arable land to mechanized agriculture; in many areas, this expanding group of nations is also experiencing forest regrowth. Conversely, stalled demographic and economic trajectories among less developed countries (LDCs) such as Honduras and Bolivia accompanied continued population growth among a largely rural (though increasingly urban) population, predominately comprised of families working in small farm, labor-intensive, and capital- and technology-poor agriculture [9,10]. Most such LDCs continued to convert old growth forests to agriculture and pastures, ever more in areas of high conservation priority [11,12]. In Central America, for example, countries undergoing the most deforestation between 2001 and 2010 were the least socio-economically developed [1]. This finding contradicts expected global diet transition impacts on forest change: that is, as countries develop, wealthier urbanizing populations demand more animal products [13], which, even under

intensively managed systems, require several times more acreage to produce the protein and caloric equivalent of vegetables and legumes [10]. This ostensible paradox may be partially explained by the disproportionate share of growing global demand for animal protein satisfied by pastoral and agricultural expansion—in the form of animal feed such as soy—within developing nations [14]. A growing body of research from local scale in-depth case studies to national and supra-national scale satellite imagery [8,15–19], has enhanced our understanding of land-use transitions at local scales. However, conclusive corroboration of interlinked impacts of global, regional, and local processes on forest change has proven elusive [3]. This study leverages a decade of change [20] to capture the local and long-distance forest change drivers simultaneously, to assess relative pushes and pulls of local and global socio-ecological and economic influences. Our research question is the following: What were the multi-scale drivers of forest cover change in LAC during the first decade of the 21st century and what were the relative influences of each?

## 2. Materials and Methods

### 2.1. Experimental Design

We constructed multilevel (hierarchical) linear regression models of annual forest change at the municipal level across Latin America and the Caribbean (LAC). Satellite imagery derived measures of annual land cover change were modelled as functions of climatic, social, economic and demographic variables derived from multiple data sources (described below), at municipal and national levels. We obtained best-fit models for deforestation and reforestation, and we describe the dependent variable parameter estimates as drivers of change at the local level. The full statistical treatment is described in the methods.

### 2.2. Data Acquisition

#### 2.2.1. Land-Cover Change Data

We build on previous work in which 250-m MODIS MOD13Q1 imagery was classified to describe land-use and land-cover trends across 16,050 municipalities in Latin America and the Caribbean (LAC) [2]. Annual forest change was estimated using maps developed from 250-m Moderate Resolution Imaging Spectroradiometer (MODIS) satellite data. Human interpretation of high-resolution imagery in Google Earth was combined with reference data for training and accuracy assessment using the Virtual Interpretation of Earth Web-Interface Tool developed by the last author to facilitate the visual interpretation of 40,432 reference samples [2]. Using a Random Forests tree-based classifier, land-use/land-cover maps were produced for LAC with eight cover classes for each year from 2001 to 2010 [2].

In our current study, we focused on the "woody vegetation" class (tree and shrub cover greater than or equal to 80% of the pixel). We refer to this class as forest in our study, and our definition includes closed-canopy forests, woodlands, and shrublands. For woody vegetation, classification accuracy was found to be as high as 98.4%, with an average accuracy of 81.8% across all 26 land-cover maps for LAC [2]. The average overall accuracy for the forest/no-forest (woody vegetation vs. all other seven classes) classification was $94.2 \pm 4.2\%$ ($n = 26$).

Our response variable was the rate of significant (at $\alpha \leq 0.05$) deforestation or reforestation from 2001 to 2010 at the municipal level, derived using the slope of a linear regression analysis of woody area (dependent) vs. time (independent, ten years) for each municipality [2,3]. We restricted our analysis to those municipalities with significant regression slope terms, allowing us to: (1) investigate factors contributing to significant change, and (2) to understand the drivers for the separate directions of forest (woody vegetation) change over the decade—gain (positive slope) and loss (negative slope). Of the 2513 municipalities with significant forest change [2,3], we include the 2233 (1305 deforestation, 928 reforestation) with data for all independent variables.

To address concerns that linear models of change might fail to capture shifting directions of forest change within the decade—for example, the reduction of deforestation rate

in the Brazilian Amazon—we examined our significant change municipalities in greater depth. First, we confirmed that municipalities with significant negative slopes had overall net forest cover loss; those with significant positive slopes had net forest gain. We then divided the annual changes across the decade into 2001–2005, and 2005–2010 and confirmed that the direction of change within each subset was consistent with overall direction of the slope of the regression for all countries in the analysis. We are therefore confident that our data supports the significant slopes modelled.

### 2.2.2. Municipal-Level Indicator Variables

Population density change from 1990 to 2000, mean elevation, mean annual precipitation, mean annual temperature and area, at the municipal scale, were examined as variables in the first level of the model. Municipal-level population density change between 1990 and 2000 was derived using national census data, as described previously [3], and represents the prior decade's increase in demand for remaining agricultural land. Temperature and precipitation data were acquired from the Climate Research Unit Datasets, University of East Anglia [21], and average elevation was derived at a 90-m resolution from the CGIAR-CSI database [22], as described previously [3].

### 2.2.3. Country-Level Indicators

We examined demographic, economic and agricultural production indicators at the national level. Total population and its rural proportion in 2000 and 2009 were acquired from FAOSTAT [23]; remittances, foreign investment and gross domestic product (GDP) in 2000 and 2009 in US dollars were obtained from World Bank Open Data [24], and 2000 and 2009 soy, beef, and corn production, in tons, were acquired from FAOSTAT [23].

### 2.3. Statistical Analysis

We constructed hierarchical models of climate and demographic change at the local (municipality) level, and demographic, agricultural production, and economic globalization trends at the national level, as simultaneous drivers of forest cover change, across LAC (Figure 1). We used multilevel models in R (version 2.13.2; packages 'lme4', 'arm', glmulti). Between and even within disciplines, diverse terms are used to describe multilevel, hierarchical, mixed effects, or nested models. For ease of explanation, we refer to this regression as a multilevel model, and describe the variance components as they appear in our specific case.

We used a basic two-level model, allowing slopes and intercepts to vary across groups, to describe the significant slope of forest cover change, $y_{ij}$, at the municipal level ($i$), within country ($j$):

$$\text{Level 1}: \; y_{ij} = \beta_{0j} + \beta_{1j}(X_{1ij}) + \beta_{2j}(X_{2ij}) + \ldots + \beta_{nj}(X_{nij}) + e_{ij}, \tag{1}$$

where $X_{1 \ldots n}$ are predictor variables at the municipal level; $e_{ij}$ is the error term subsuming the independent error for the intercept $\beta_0$ and the independent error of the regression coefficients $\beta_1$ to $\beta_n$, and the predictors $X_1$ to $X_n$. $\beta_0$ to $\beta_n$ are the regression coefficients, whose variation depends on explanatory variables at the country level, for example:

$$\text{Level 2}: \; \beta_{0j} = \gamma_{00} + \gamma_{01}(Z_j) + \mu_{0j}, \tag{2}$$

in which $\gamma_{00}$ is the intercept for the overall model of $\beta_0$, and $Z_j$ is the country-level predictor; with the residual error $\mu_j$ at the country level.

As we had multiple municipal and country level predictors, this can be summarized with $X$ taking subscript $p$ ($1 \ldots P$), and $Z$ taking $q$ ($1 \ldots Q$), as:

$$Y_{ij} = \gamma_{00} + \sum_p \gamma_{po} X_{pij} + \sum_q \gamma_{0q} Z_{qj} + \sum_p \sum_q \gamma_{pq} X_{pij} Z_{qj} + \sum_p \mu_{pj} X_{pij} + \mu_{0j} + e_{ij}. \tag{3}$$

We also examined the effect of state ($k$) and terrestrial biome ($l$) [25] as additional levels explaining the data structure. An advantage to using hierarchical models is that

coarser-scale levels can help control for spatial autocorrelation among the municipalities, the finest scale. As we include no explanatory variables at the state and biome levels, these appeared as part of the variance, which we present as a modification of Equation (3):

$$Y_{ijkl} = \gamma_{0000} + v_{0k} +_{0l} + \ldots + \mu_{0jkl} + e_{ijkl}. \tag{4}$$

Thus, $\sigma^2 v_{0k}$ is variance at the state level and $\sigma^2 o_{0l}$ is variance at the terrestrial biome level.

As our hypotheses included predictors that are likely to be correlated, we mean-centered all variables and examined variance inflation factors (VIFs) of the parameters, and kappa statistics for collinearity effects on the overall model. For all predictor variables, we used proportional changes in the indicators, and centered the variable; at the municipal level, we used the data as presented above, centered. Centering predictors assists interpretation in multilevel models by allowing examination of relative change on the mean (average) property of a level at the higher level. Additionally, it helps reduction of collinearity effects on estimates and tends to improve model convergence.

We conducted predictor selection by examining the variables used at each level in a multi-model comparison using Akaike's Information Criterion (AIC) to select the best candidate set of variables [26], using the R package 'glmulti' [27]. We did this for deforestation (significant negative slopes in woody area vs. time, in municipality-level regression models) and reforestation (significant positive slopes) separately. The resulting candidate variable sets (given in Table 1), in combination with the structural effects of biome and state, were used in the multilevel models, allowing all slopes and intercepts to vary.

**Table 1.** Parameter estimates, and standard errors (SE) for best-fit models of deforestation and reforestation in LAC, excluding state- and biome-level predictor variables. Significant variable estimates are in bold.

| | *Estimate* | *SE* | | *Estimate* | *SE* |
|---|---|---|---|---|---|
| **Deforestation** | | | **Reforestation** | | |
| Intercept | −12.61 | 24.76 | **Intercept** | **7.12 \*\*\*** | **2.05** |
| *Municipal Level* | | | *Municipal Level* | | |
| **Temperature** | **−4.41 \*\*\*** | **1.47** | Temperature | 0.28 | 0.20 |
| Precipitation | 0.00 | 0.00 | Precipitation | −0.01 | 0.01 |
| **Municipality Area** | **−0.01 \*\*\*** | **0.00** | **Municipality Area** | **0.01 \*\*\*** | **0.00** |
| *Country Level* | | | *Country Level* | | |
| Soy Production | 0.83 | 1.37 | **Soy Production** | **−2.72 \*\*\*** | **0.44** |
| **Beef Production** | **−61.32 \*\*** | **22.97** | **Foreign Investment** | **−0.65 \*** | **0.29** |
| Foreign Investment | −0.39 | 0.81 | Population Change | −18.02 | 23.55 |
| **Remittances** | **−2.51 \*** | **1.22** | **GDP** | **−6.29 \*\*** | **2.00** |
| Population Change | 86.58 | 74.62 | | | |
| Rural Proportion | −192.80 | 227.90 | | | |
| *Quasi $R^2$* | *0.51* | | *Quasi $R^2$* | *0.68* | |

\* Significant at $p = 0.05$; \*\* Significant at $p = 0.01$; \*\*\* Significant at $p < 0.001$.

The base (or intercept only) model $y_{ij} = \beta_{0j} + e_{ij}$, was used to establish the structure accounted for in the data at the country level, to compare the impact of adding predictors at the two levels (municipality ($i$) and country ($j$)). We created baseline models for negative slopes (deforestation) and positive slopes (reforestation), and derived AIC values, using maximum likelihood estimation in R. We then stepped through three stages of predictor and factor addition: adding the predictors at both levels, adding 'state', and adding biome. In each stage, model improvement over the previous was assessed, with the criteria of 'improvement' at $\Delta$AIC $\geq 2$ [26].

For ease of interpretation of the best model fits, we assessed the significance of parameters using t-tests, assuming that our large sample sizes (928 and 1305) and relatively few estimated parameters (10 and 8) increased certainty about estimating degrees of freedom (DF), which would exceed 500, often the point of reported convergence of the critical value

at 1.96 for $\alpha = 0.05$. We also constructed quasi-$R^2$ measures of model fit as the $R^2$ from a linear regression of the predicted and observed model, obtained from R package 'lme4', recognizing that mixed model structures do not lend themselves to true $R^2$ values.

Previous work [3] found that the terrestrial biome, as defined by Olson et al. [25] was a significant predictor of municipal-level forest change. More than 80% of deforestation occurred in moist forest, dry forest, and savannah/shrubland biomes, suggesting that these areas are most vulnerable to the conversion of forest to agriculture, rather than to other land cover transitions. Conversely, reforestation occurred largely in xeric shrubland systems [2,3]. We included both of these structural variables (biome and state) as factors in models (see Equation (4)) and assessed model fit improvement.

## 3. Results

This analysis spans 14 countries, covering 92% of LAC's land area (Figure 1). Using a hierarchical linear regression framework (see Methods) we simultaneously assessed the strengths of proportional changes in local and national level pressures on annual municipal ($n = 16{,}050$) forest cover change during 2001–2010, and we explored results relative to global agricultural trade.

Deforestation and reforestation occurred simultaneously across LAC from 2001 to 2010. Agricultural expansion [28] and intensification [29,30] in high production regions were accompanied by reforestation and depopulation in some less productive regions [31]. The inclusion of state-level effects accounted for additional variance in the data, while biome-level inclusion did not, suggesting the importance of state-level socio-economic and political variation in land use and land cover change.

Deforestation, with beef production its largest significant driver, far exceeded reforestation (Figure 2, Table 1). A doubling in national beef production over the decade was associated with a mean of 61.32 km$^2$ municipal-level deforestation.

Predicted municipal-level deforestation area (km$^2$) resulting from national-level production or economic change (2001–2010) of the four largest significant model predictors of deforestation and reforestation (beef, soy, remittances, GDP) for the 14 countries in this analysis.

During the decade, Argentina, Brazil, Uruguay and Paraguay (Figure 3a,b) were among the top ten beef exporters. Brazil ranked number one, shipping abroad nearly four times that of the European Union (EU-27) (Figure 3b). The Russian Federation was the largest importer of LAC beef (Figure 3b); the U.S., the largest importer of LAC live cattle, most arriving from neighboring Mexico [23].

The destination and quantity in tons of a. beef (cattle meat, beef, and veal (boneless)), not including live cattle, and c. soy (soybeans, soy cake), exported by the top five LAC exporting countries to the top ten importing countries, 2000–2010; and the proportional contributions of LAC exports to the top ten importing countries from 2000 to 2010 of b. beef (inset of production quantity of top 5 LAC countries) and d. soy (inset production quantity of top 5 LAC countries) [Detailed trade data from FAOSTAT (nd)].

National GDP had the largest impact on reforestation (Table 1). A 6.29 km$^2$ reduction in local forest gain corresponded to each doubling of per capita GDP. Foreign investment at the national level also reduced reforestation. Municipality size related positively to modest forest regrowth. As meat consumption typically rises with disposable income [13], the large effect of national GDP on slowing reforestation supports the observed association between LAC beef production and deforestation. Similar to the deforestation findings, national-level population change and mean annual municipal precipitation and temperature contributed to the best fit reforestation model. National soy production was the largest significant agricultural factor mitigating reforestation. Halving soy production over the decade was associated with an average municipality forest cover gain of 2.72 km$^2$. From 2000 to 2010, LAC was the global leader in soy exports, led by Argentina and Brazil (Figure 3c,d); China was the top global importer of soybeans, shifting from the equivalent of the EU-27 nations to three times EU imports by 2010 (Figure 3d). Argentine and Brazilian exports comprised the majority of China's LAC soy imports [23].

Our combined deforestation and reforestation analyses suggest that pasture expansion and mounting soybean production, in response to increasing global demand for food and feed [32,33], were the predominant drivers of recent LAC deforestation, a trend saliently exemplified in the southern Brazilian Amazon [34]. Municipality size was a significant positive predictor of both reforestation and deforestation, suggesting continued forest cover dynamism in frontier regions. We observed only modest evidence for local population impacts on forest change, corroborating related research [35]. While LAC retains global leadership in beef production, exporting to nations of high (EU) or rising (Russia) affluence, its own increasingly prosperous and urban populations consume the majority of its beef. Conversely, a majority of its soy exports has fed poultry and swine abroad. Approximately 2% of global soy is directly consumed by humans; 98% is processed for soymeal to feed livestock [36]. An estimated 85% of this feed is destined for poultry and pig production [37], together representing most of China's doubled meat consumption since 1980. LAC's largest soy importer, China, now consumes approximately one-third of global meat production [23].

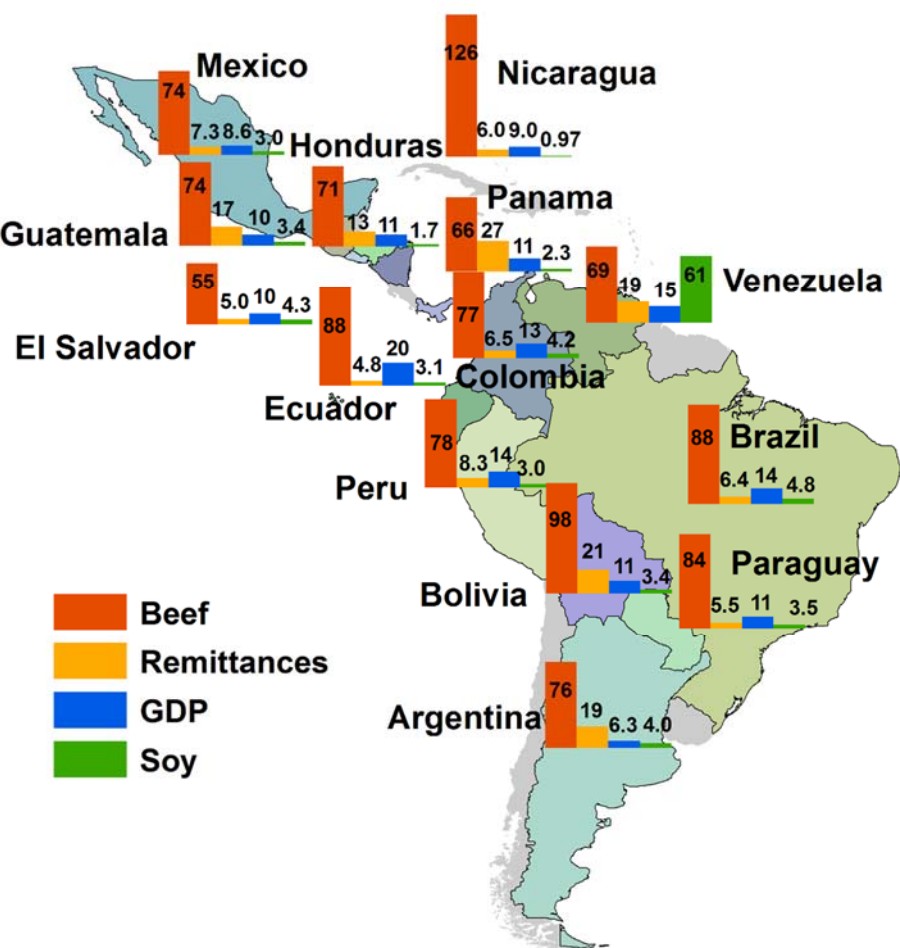

**Figure 2.** The four largest national level predictors of municipal-level deforestation in LAC (2001–2010).

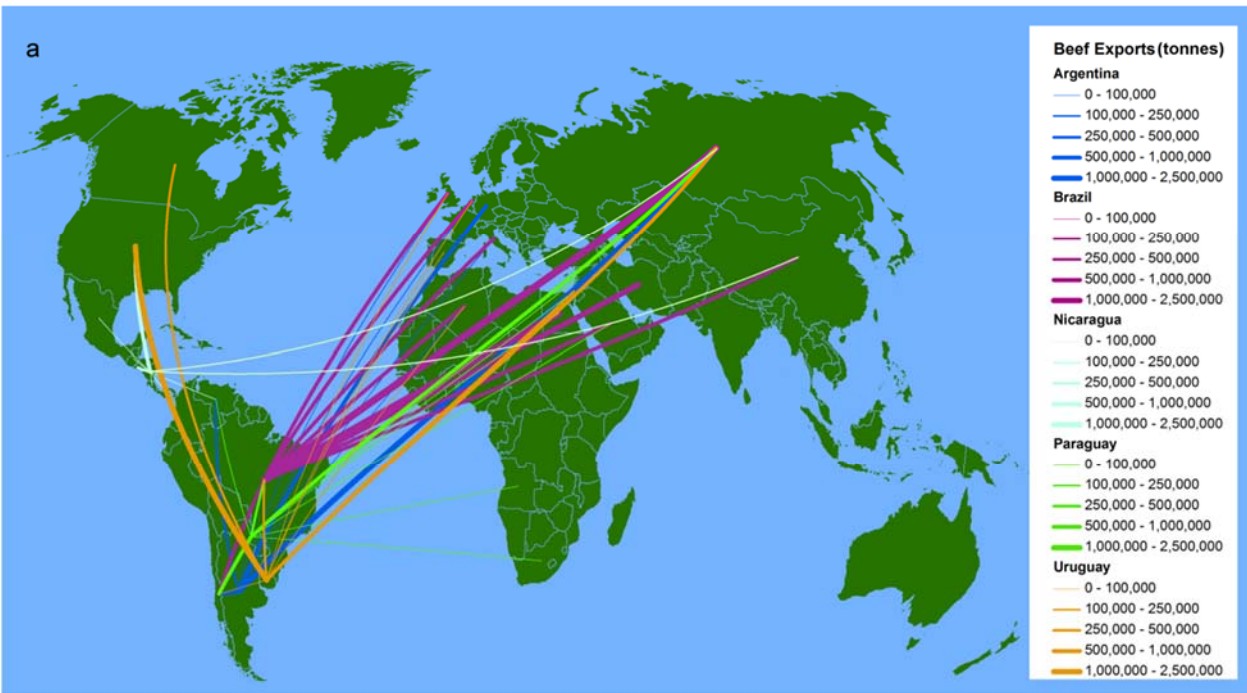

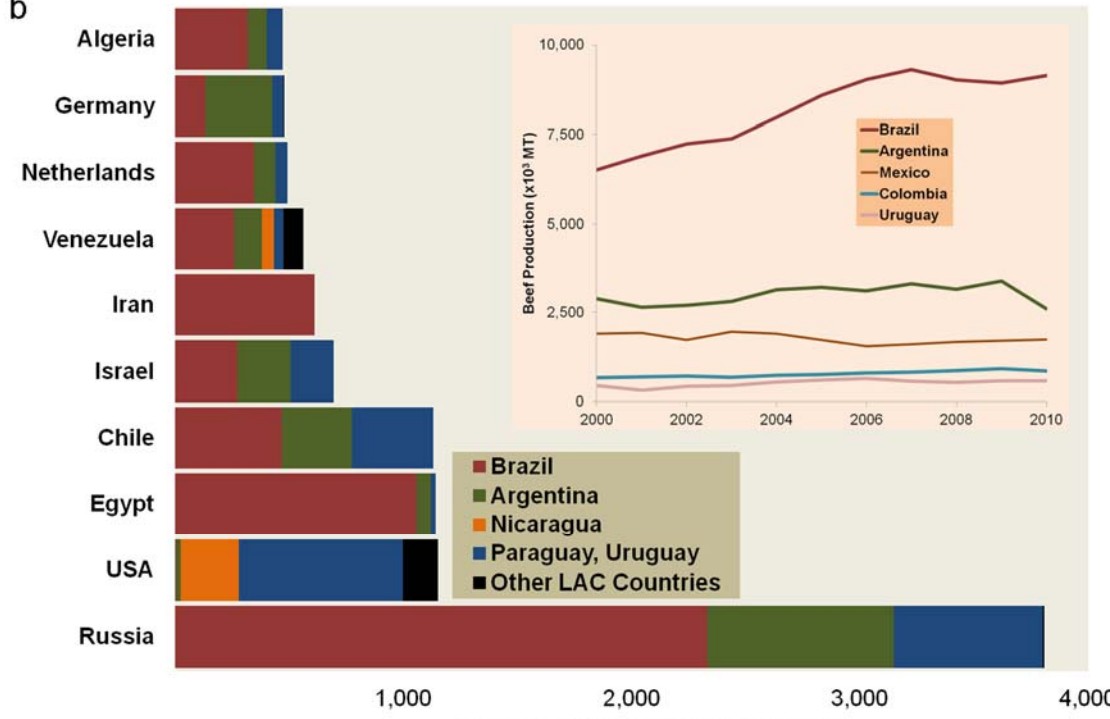

**Figure 3.** *Cont.*

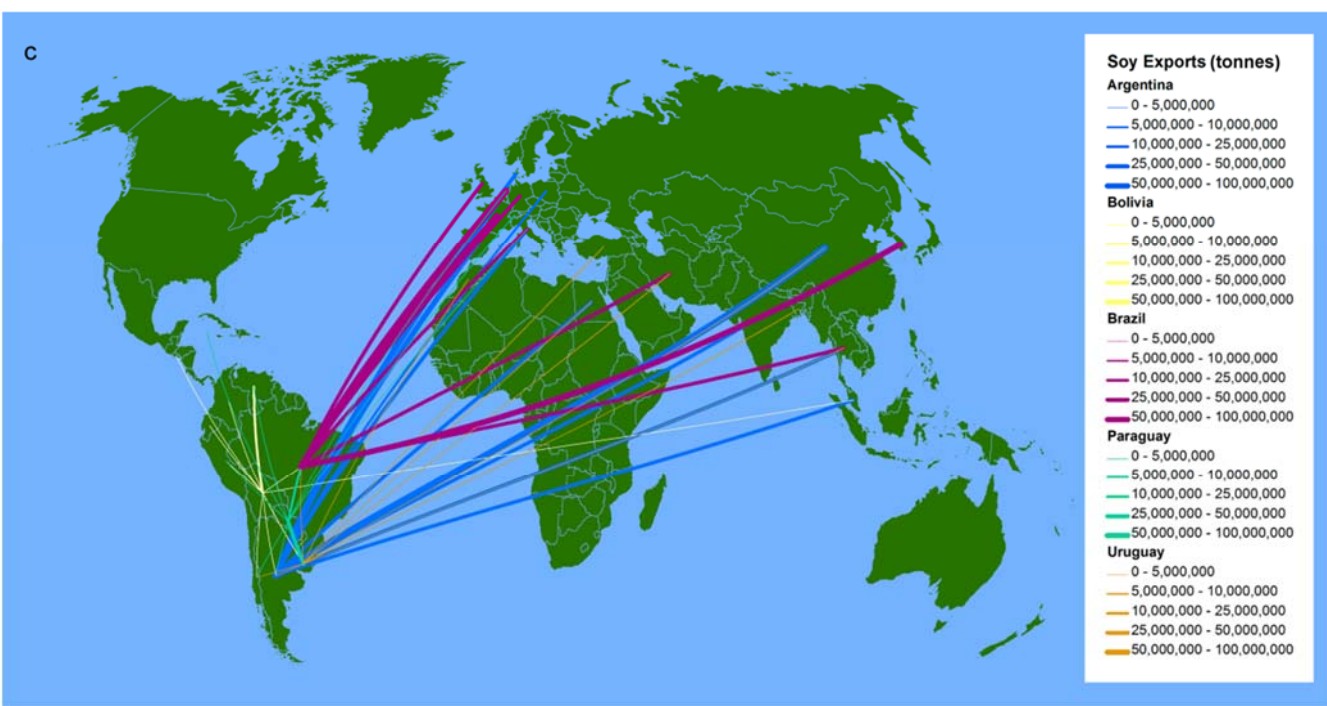

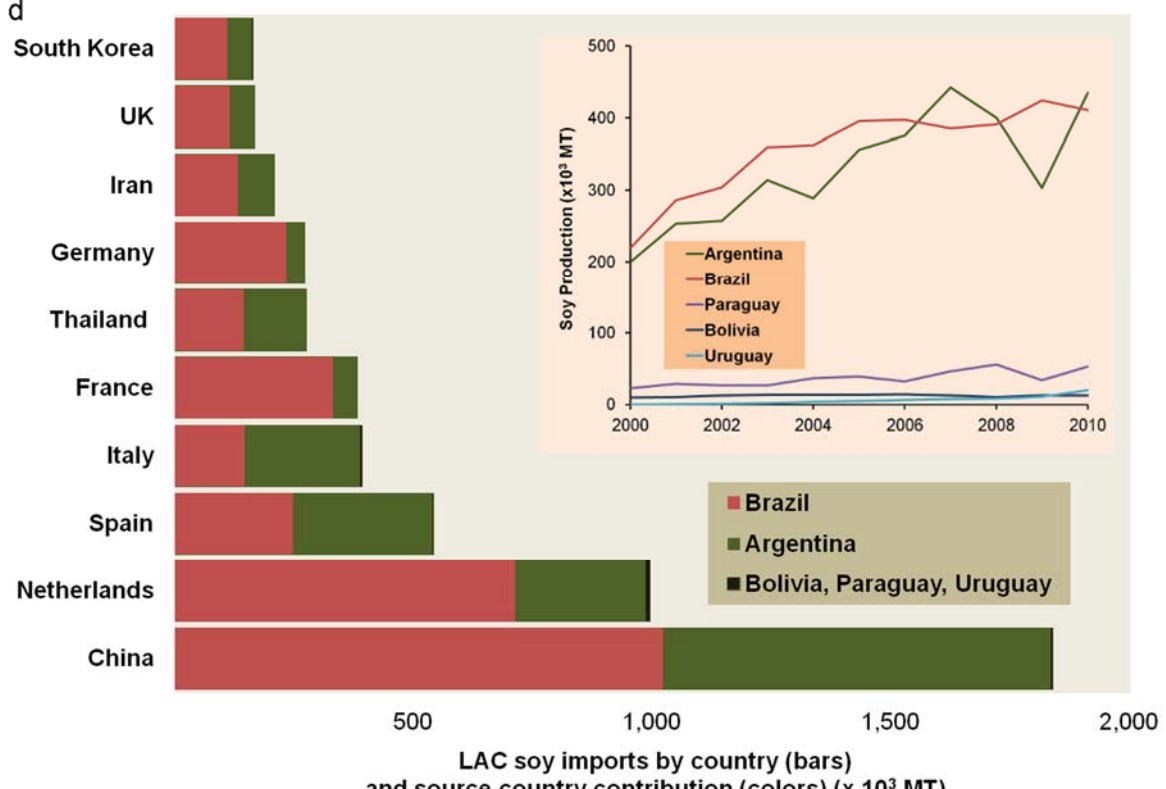

**Figure 3.** (**a–d**) Trade flows and production for beef and soy from LAC to top importers 2000–2010.

## 4. Discussion

LAC deforestation is of great concern for sustainable human development and forest conservation. More than a third of global deforestation in 2001–2010 occurred in LAC [2]. Current available LAC agricultural land may become exhausted as early as the late 2020s [4], likely necessitating dramatic production increases from currently forested

areas. Our findings support the importance of modelling land change across local, national, and international scales. When integrating local to international level processes, our findings challenge forest transition theories, which anticipate that countries further along the development gradient should be associated with rural depopulation, intensified agricultural production, and increasing reforestation [1,8]. Instead, our analysis indicates that an interconnected world blurs the line between developing and developed regions. Results suggest that a rising consumer class, regionally and abroad, has swelled demand for meat and dairy products, spurring agricultural and pastoral expansion at the expense of existing forests. Specifically, during the 1st decade of the 21st century, regional forest change appeared to be driven by LAC's global leadership in beef and soy exports to meet international demand for animal protein as well as by increased domestic consumption facilitated by a growing consumer class buoyed by rising GDPs and by capital inflows from migrant remittances and foreign investment. These drivers of considerable deforestation in LAC may also partially explain some recent reforestation observed in MDCs to the extent these nations meet growing domestic demand for animal protein by exporting their agricultural expansion to LAC [18]. The exigencies of multinational economic integration and mounting local and global demand for animal protein, accompanied by rising regional affluence and urbanization, explained much of the dramatic LAC deforestation observed during the first decade of the 21st century. Such global and local, integrated, multi-scale processes must be examined ensemble in future research in order to properly inform local, national, and international ecological and socio-economic policy related to land change. More spatially nuanced conceptual and methodological approaches are important if we are to reconcile regional forest conservation with surging global demand for animal protein within a context of accelerating climate change.

**Author Contributions:** S.J.R. performed statistical analyses; S.J.R. and D.L.-C. interpreted results. S.J.R., D.L.-C. and M.L.C. wrote the paper. Data provided by D.L.-C. and M.L.C. All authors have read and agreed to the published version of the manuscript.

**Funding:** This work was supported by funding from the National Science Foundation Dynamics of Coupled Natural and Human Systems (NSF CNH EF 0709627 and 0709645) to D.L.-C. and M.L.C., and S.J.R.'s postdoctoral work by the National Center for Ecological Analysis and Synthesis, a Center funded by NSF (Grant #EF-0553768), the University of California, Santa Barbara, and the State of California.

**Institutional Review Board Statement:** Not applicable.

**Informed Consent Statement:** Not applicable.

**Data Availability Statement:** Not applicable.

**Acknowledgments:** We thank Jarrett Byrnes for his input on statistical interpretation and figure construction.

**Conflicts of Interest:** The authors declare no conflict of interest.

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
