# Peer review of "Global Economic and Diet Transitions Drive Latin American and Caribbean Forest Change during the First Decade of the Century: A Multi-Scale Analysis of Socioeconomic, Demographic, and Environmental Drivers of Local Forest Cover Change"

_land, doi:10.3390/land11030326_

Round 1

Reviewer 1 Report

This is an interesting paper, opening a better understanding about drivers of forest change! 

For me, the paper's title reads like a heading of any newspaper. In fact, it presents a new concept an in-depth analysis of drivers of deforestation, linking impacts from global trade to local changes in forest cover. In a scientific journal, this should come out of the title. Consider renaming the title. 

If you agree that we are dealing with a research paper, I would expect a more or less detailed set of objectives, research gaps and research questions etc.. The text starting from Line 67 seems to be too general for me.

Line 119: How did you synchronize data from 1990-2000 with the period 2000-2010? 

Line 197ff: How do you include "Biome" in that model and what about "State"

Fig. 1: There are some gaps in the colorscale. Does it show the balance of gain and loss ?

Fig. 2: I am not a native English speaker, but for me the word predictor is not suitable in this case. I would say your variables are "estimators" (??). Can you add a unit to the columns?

Tab. 1: What do you mean with SEM? SE?

After showing Table 1 which presents the results of your analysis parameters of the model, you start with linking the global trade data with national and local data. That is ok, but for better understanding and better readability it can be helpful starting a new subchapter within the results chapter. Here you give interesting examples how to use the model and how it can be used to attribute global trade to forest change 

Author Response

Reviewer 1

This is an interesting paper, opening a better understanding about drivers of forest change! 

For me, the paper's title reads like a heading of any newspaper. In fact, it presents a new concept an in-depth analysis of drivers of deforestation, linking impacts from global trade to local changes in forest cover. In a scientific journal, this should come out of the title. Consider renaming the title. 

--Thank you for this recommendation. We have added the following to the title to better reflect the contents of the paper: A multi-scale analysis of socioeconomic, demographic, and environmental drivers of local forest cover change  

If you agree that we are dealing with a research paper, I would expect a more or less detailed set of objectives, research gaps and research questions etc.. The text starting from Line 67 seems to be too general for me.

--We appreciate this important comment. Following this comment, we have added the following text to the last paragraph of the introduction:

However, conclusive corroboration of interlinked impacts of global, regional, and local processes on forest change has proven elusive [3]. This study leverages a decade of change [20] to capture the local and long-distance forest change drivers simultaneously, to assess relative pushes and pulls of local and global socio-ecological and economic influences. Our  research question is the following: What were the multiple scale drivers of forest cover change in LAC during the first decade of the 21st century and what were the relative influences of each?

Line 119: How did you synchronize data from 1990-2000 with the period 2000-2010? 

--As described in this methods section, the prior decade’s increase in population density was used to describe the pressure remaining agricultural landscapes. We did not mean this to be synchronized (which I’m interpreting as simultaneous timing) with the cover change in question. To avoid issues of collinearity, we used population metrics that the national level during the time period, which did not describe density, rather other characteristics.

Line 197ff: How do you include "Biome" in that model and what about "State"

--This is indicated at line 168 in describing equation equation 4, in which state (k) and terrestrial biome (l) are included as part of the variance (see eqn 4).

We have updated the language to indicate this to the reader more clearly in the identified line (reviewer line 197ff, our copy line 211):

We included both of these structural variables (biome and state) as factors in models (see eqn 4.) and assessed model fit improvement.

Fig. 1: There are some gaps in the colorscale. Does it show the balance of gain and loss ?

--We are not clear what the reviewer is asking – this is a dichromatic green-red scaled map, with 10 classes, and the inflection around zero in class 5. The class breaks are set manually at 0.5, 10, 100, 500, to reflect the km2 of forest cover loss at the municipal level for all of the significant change municipalities across the 14 countries in the analyses. If the figure colors are not rendering in the reviewer copy, and gaps in the colors are occurring, we request editor assistance with the figure itself. 

Fig. 2: I am not a native English speaker, but for me the word predictor is not suitable in this case. I would say your variables are "estimators" (??). Can you add a unit to the columns?

--We respectfully disagree; in regression models, the variables are predictors; however, for the table, we updated the legend to say ‘significant estimates are in bold’.

Tab. 1: What do you mean with SEM? SE?

--We updated this to SE, thank you.

After showing Table 1 which presents the results of your analysis parameters of the model, you start with linking the global trade data with national and local data. That is ok, but for better understanding and better readability it can be helpful starting a new subchapter within the results chapter. Here you give interesting examples how to use the model and how it can be used to attribute global trade to forest change 

--We appreciate this comment but the global trade data only accounts for a few lines that are used to help contextualize our model results. It would be awkward to have a separate section just for a few lines of text.

Thank you for your comments. We believe that they have helped us improve the paper.

Reviewer 2 Report

The authors submitted a nice and easy to read MS, so my comments are positive. My main concern is about the equations 1-3? It seems to me that, as they are, they do not work.  If at level 2, only Beta_0j varies with Z, than eq. 3 should follow y_ij=Beta_0j(Z_j)+Beta_2j(X_ij)+…. . Based on Eq. 1., it is neither clear whether Beta_ij(X_ij) means that Beta is a function of X_ij, or whether it is multiplied with X_ij. Because the authors stated earlier that a linear approach was employed, and because of the eq. 2, I assume that my second guess is correct. In this case, I would recommend that the authors leave the brackets from the e1q. 1. If eq. 3 is correct, I guess that an index should replace the symbol ‘0’ in eq. 2. If I am mistaken, then the gamma_00 in eq. 3 should be replaced with (n+1)gamma_00, and so symbol mu should be summed across all mu from eq. 2.  The Y (capital) in eq. 3 is the same as the y (small letter) in eq. 1, is not it? At line 178, the authors state that beta_0j+error term is an intercept, but earlier they (in my opinion correctly) state that gamma_00 is the intercept.

I hope that these inconsistencies are only it the presentation and the equations were inputted correctly to the computational tool. If these problems occurred on the SW input, the results might be biased.

Some other comments are very minor indeed.

Line 15. Does ‘more than a third of…’ means ‘more events than the third …’ or ‘more than three global …’, or something else?

  1. 23 is my reading that the area of municipalities has an effect on percentage of deforestration correct?

  1. 42-43 hard to follow

  1. 45. The sentence commencing with ‘More’ and terminating at line 49 is very hard to follow

  1. 77 ‘… information [?from?] theoretical approaches…’. I would appreciate if the authors provided some details about the approaches.

  1. 78. The expression ‘Best fit models… ’ sounds me as if the models were not linear, because the authors picked up the best model from a pool of models. Based on further reading, I guess that the authors picked up from pool of variables (employing AIC) and the model was linear (the only model was linear so there were not best and worse models). Is that correct? If so, the sentence deserved to be improved.

  1. 88 ‘web-based tool’, please, provide a reference.

  1. 95 ‘pixel-level accuracy’, please specify

  1. 83-91 I would appreciate if the authors expanded the whole paragraph. As it is, it reads cryptic to me. The reader may not have the paper [2] at the top of their finger.

  1. 159. I read ‘no explanatory’ variables as those with insignificant effect. Based on the further reading I think that the authors mean the variables excluded by AIC analysis. What is correct? Please, would you specify?

  1. 163. Does the expression ‘mean-centered’ means ‘with zero sum of errors’?

Do I read the paper correctly, if my impression is that the soy production is insignificant? If so, it contradicts me with the scenario that soy is fed by cows in less ‘developed’ countries. ls. 49-63

If my reading is correct I would appreciated that the authors discuss their finding more visibly, as it sounds me novel and contra intuitive. If I am mistaken, ith would be worth improving the MS in order to avoid this mistake for other readers.

How the authors classify more and less developed countries? Would not be more accurate classify them along an axis of GDP, of percentage of urban population?, to offer two possibilities.

I hope that my comments will help the authors to improve their MS.

Author Response

Reviewer 2

The authors submitted a nice and easy to read MS, so my comments are positive. My main concern is about the equations 1-3? It seems to me that, as they are, they do not work.  If at level 2, only Beta_0j varies with Z, than eq. 3 should follow y_ij=Beta_0j(Z_j)+Beta_2j(X_ij)+…. . Based on Eq. 1., it is neither clear whether Beta_ij(X_ij) means that Beta is a function of X_ij, or whether it is multiplied with X_ij. Because the authors stated earlier that a linear approach was employed, and because of the eq. 2, I assume that my second guess is correct. In this case, I would recommend that the authors leave the brackets from the e1q. 1. If eq. 3 is correct, I guess that an index should replace the symbol ‘0’ in eq. 2.

--We apologize for the inconsistency in describing the regression components in the system of equations between equations 1 and 2 – there are indeed parentheses missing in gamma01Zj – it should read gamma01(Zj) – we have updated equation 2 in the manuscript to reflect this.

We did not provide the full expansion of the second level equations for the purposes of space; the equation 2 expansion of beta0j is the example, as stated on our line 162.

If I am mistaken, then the gamma_00 in eq. 3 should be replaced with (n+1)gamma_00, and so symbol mu should be summed across all mu from eq. 2.  The Y (capital) in eq. 3 is the same as the y (small letter) in eq. 1, is not it? At line 178, the authors state that beta_0j+error term is an intercept, but earlier they (in my opinion correctly) state that gamma_00 is the intercept. I hope that these inconsistencies are only it the presentation and the equations were inputted correctly to the computational tool. If these problems occurred on the SW input, the results might be biased.

--We shifted the notation for the multilevel model from systems of equations model layout to a mixed-effects model notation, between equations 2 and 3, to express it more compactly. In the level 1 equation, beta0j + error is an intercept, but when we summarize the whole system into the mixed-effects format, gamma 00 is the notation for the intercept.

The models themselves were inputted correctly, and it is, as the reviewer suggests, the expression of the models as equations, and which notation we used (and a missing set of parentheses) that seem to be the issue here. 

Some other comments are very minor indeed.

Line 15. Does ‘more than a third of…’ means ‘more events than the third …’ or ‘more than three global …’, or something else?

--Thank you for the question. It means that of all the global deforestation occurring during the first decade of the 21st century, LAC accounted for 1/3 of this global deforestation.

  1. 23 is my reading that the area of municipalities has an effect on percentage of deforestration correct?

 --Yes, correct.

  1. 42-43 hard to follow

 --Thank you. We have clarified that the agricultural expansion is driven by deforestation.

  1. 45. The sentence commencing with ‘More’ and terminating at line 49 is very hard to follow

 --Thank you for the comment. We have divided this sentence into two separate clauses to help clarify.

  1. 77 ‘… information [?from?] theoretical approaches…’. I would appreciate if the authors provided some details about the approaches.

-- Thank you for noting this. The sentence did not make sense and we deleted the part about theoretical approaches.

  1. 78. The expression ‘Best fit models… ’ sounds me as if the models were not linear, because the authors picked up the best model from a pool of models. Based on further reading, I guess that the authors picked up from pool of variables (employing AIC) and the model was linear (the only model was linear so there were not best and worse models). Is that correct? If so, the sentence deserved to be improved.

--Multilevel regression models of this type are hierarchical linear regression models. We did not allow any of the variable responses to take on non-linear (e.g. curvilinear or polynomial) response shapes, so these are linear regression models. Using information theoretical approaches – using AIC to assess model fit improvement, we compared many of these models in an unbiased way, to arrive at the best-fit model for each direction of forest cover change.

I respectfully suggest that the reviewer is interpreting ‘linear’ in a way not intended here, and that re-wording the sentence is not warranted.

  1. 88 ‘web-based tool’, please, provide a reference.

 --The reference is provided:

Clark ML, Aide TM, Riner G. Land change for all municipalities in Latin America and the Caribbean assessed from 250-m MODIS imagery (2001–2010). Remote Sensing of Environment. 2012;126: 84–103.

  1. 95 ‘pixel-level accuracy’, please specify

--Pixel-level is not necessary in the sentence and we have deleted it. The idea is that it is at the level of the remote sensing resolution.  

  1. 83-91 I would appreciate if the authors expanded the whole paragraph. As it is, it reads cryptic to me. The reader may not have the paper [2] at the top of their finger.

-- Thank you for this comment. While we do not wish to repeat the whole methodology of the prior study we have added a couple of additional details by way of explanation.

  1. 159. I read ‘no explanatory’ variables as those with insignificant effect. Based on the further reading I think that the authors mean the variables excluded by AIC analysis. What is correct? Please, would you specify?

--I believe the reviewer is referring to what appears in my line 165 in the ms, in which we explain that we include the state and biome without drivers at those levels. There are no explanatory variables in the model at those levels. Only levels. They are part of the variance, as shown in equation 4.

  1. 163. Does the expression ‘mean-centered’ means ‘with zero sum of errors’?

--Mean centering the variables prior to modeling involved subtracting the mean value of the variable from each value, thus retaining the magnitude of direction of the driver (and interpretation), while allowing model convergence.

Do I read the paper correctly, if my impression is that the soy production is insignificant? If so, it contradicts me with the scenario that soy is fed by cows in less ‘developed’ countries. ls. 49-63

--This passage was in reference to Central America, where soy is not a major crop (unlike South America). 

If my reading is correct I would appreciated that the authors discuss their finding more visibly, as it sounds me novel and contra intuitive. If I am mistaken, ith would be worth improving the MS in order to avoid this mistake for other readers.

--Thank you for the comment. We have tried to explain the findings as visibly as possible with the following text:

Results suggest that a rising consumer class, regionally and abroad, has swelled demand for meat and dairy products, spurring agricultural and pastoral expansion at the expense of existing forests. Specifically, during the 1st decade of the 21st century, regional forest change appeared to be driven by LAC’s global leadership in beef and soy exports to meet international demand for animal protein as well as by increased domestic consumption facilitated by a growing consumer class buoyed by rising GDPs and by capital inflows from migrant remittances and foreign investment. 

How the authors classify more and less developed countries? Would not be more accurate classify them along an axis of GDP, of percentage of urban population?, to offer two possibilities.

 --Thank you for this question. We are not classifying them. Rather we are discussing them as on a continuum from less to more developed.

I hope that my comments will help the authors to improve their MS.

--Thank you for your helpful review. We believe the paper is now improved thanks to your and reviewer 1’s comments.